# Endovascular Electroencephalogram Records Simultaneous Subdural Electrode-Detectable, Scalp Electrode-Undetectable Interictal Epileptiform Discharges

**DOI:** 10.3390/brainsci12030309

**Published:** 2022-02-24

**Authors:** Ayataka Fujimoto, Yuji Matsumaru, Yosuke Masuda, Aiki Marushima, Hisayuki Hosoo, Kota Araki, Eiichi Ishikawa

**Affiliations:** 1Comprehensive Epilepsy Center, Seirei Hamamatsu General Hospital, Shizuoka 988-056, Japan; afujimotoscienceacademy@gmail.com; 2School of Rehabilitation Sciences, Seirei Christopher University, Shizuoka 988-056, Japan; 3Department of Neurosurgery, Faculty of Medicine, University of Tsukuba, 1-1-1, Tennodai, Tsukuba 305-8575, Japan; y-masuda@sis.seirei.or.jp (Y.M.); aiki.marushima@md.tsukuba.ac.jp (A.M.); hisayuki.hosoh@gmail.com (H.H.); araki.kota.lp@alumni.tsukuba.ac.jp (K.A.); e-ishikawa@md.tsukuba.ac.jp (E.I.); 4E.P. Medical Inc., 403 Nihonbashi-Life-Science Building, 2-3-11, Honcho, Nihonbashi, Chuo-ku, Tokyo 103-0023, Japan

**Keywords:** endovascular electroencephalogram, localized epileptogenicity, microcatheter, focal onset epilepsy, interictal epileptiform discharge

## Abstract

Introduction: We hypothesized that an endovascular electroencephalogram (eEEG) can detect subdural electrode (SDE)-detectable, scalp EEG-undetectable epileptiform discharges. The purpose of this study is, therefore, to measure SDE-detectable, scalp EEG-undetectable epileptiform discharges by an eEEG on a pig. Methods: A pig under general anesthesia was utilized to measure an artificially generated epileptic field by an eEEG that was able to be detected by an SDE, but not a scalp EEG as a primary outcome. We also compared the phase lag of each epileptiform discharge that was detected by the eEEG and SDE as a secondary outcome. Results: The eEEG electrode detected 113 (97%) epileptiform discharges (97% sensitivity). Epileptiform discharges that were localized within the three contacts (contacts two, three and four), but not spread to other parts, were detected by the eEEG with a 92% sensitivity. The latency between peaks of the eEEG and right SDE earliest epileptiform discharge ranged from 0 to 48 ms (mean, 13.3 ms; median, 11 ms; standard deviation, 9.0 ms). Conclusion: In a pig, an eEEG could detect epileptiform discharges that an SDE could detect, but that a scalp EEG could not.

## 1. Introduction

The electroencephalogram (EEG), dense array EEG, and magnetoencephalogram (MEG) are available as techniques to measure neural interactions at the network level.

These techniques are utilized to allow an indirect estimation of the epileptogenic zone for presurgical evaluations of patients with antiseizure medication (ASM)-resistant intractable epilepsy [1]. In terms of indirect techniques to estimate the epileptogenic zone presurgically, brain magnetic resonance imaging (MRI) [2,3] and other evaluations, including nuclear imaging [4], are also important. These techniques represent the gold standard as presurgical evaluation tools. However, in defining approaches to the epileptogenic zone, the major obstacles for these modalities are the indirect nature of measurements and differences from the depth or subdural electrode (SDE) recordings. The algorithm for modern epilepsy surgery, therefore, shows a stereotactic EEG (SEEG) or SDE following indirect techniques such as an EEG, dense array EEG, MEG, and MRI [5]. However, direct recordings require invasive procedures such as a skull penetration or craniotomy, which occasionally lead to adverse effects [6]. Although some studies have reported SEEGs to be less invasive and safer than SDEs [7,8], the ideal method to detect epileptic foci would be a noninvasive, direct measurement without a need for a craniotomy or penetration of the skull.

One step towards making this concept a reality could be an endovascular EEG (eEEG). This endovascular approach was introduced in the 1970s [9]. In 1995, Nakase et al. [10] performed an eEEG simultaneously with SDE recordings for a patient. From these studies, the eEEG was found to detect neural activities. García-Asensio et al. [11] provided the first statistical analysis of the utility of eEEG, showing a 93% sensitivity and 80% specificity [12]. This high sensitivity of the eEEG was supported by a study showing that epileptiform discharges undetectable by a scalp EEG were detectable by the eEEG [13]. In addition, artificially created epileptiform discharges were detected by the eEEG in an animal experiment [14].

Past studies into the eEEG have found that: (1) an eEEG can detect epileptiform discharges; (2) an eEEG offers a higher sensitivity than a scalp EEG [11,12]. However, these results were from independent eEEG studies with or without simultaneous SDE studies, and no studies simultaneously using an eEEG, SDE and scalp EEG have been reported.

We hypothesize that an eEEG could detect SDE-detectable, scalp EEG-undetectable epileptiform discharges. The purpose of this study is, thus, to measure SDE-detectable, scalp EEG-undetectable epileptiform discharges by an eEEG with a simultaneous recording in a pig.

## 2. Methods

This study was carried out in strict accordance with the recommendations in the Guide for the Care and Use of Laboratory Animals of the National Institutes of Health. The protocol was approved by the Institutional Animal Care and Use Committee of Nihon Bioresearch Inc. (study number 410028). All efforts were undertaken to minimize suffering in the male domestic pig (Gottingen Miniature pig) utilized in this study.

### 2.1. Materials

A male, 21.2 kg domestic pig under general anesthesia was utilized in this study to artificially generate an epileptic field [14,15] that was detectable by SDE, but not by scalp EEG, using detection on eEEG as the primary outcome. We also compared the phase lag of each epileptiform discharge detected by eEEG and SDE as a secondary outcome.

#### 2.1.1. General Anesthesia

The pig was fasted for 12 h before the induction of general anesthesia. Midazolam (Dormicum^®^, 1.5 mg/kg) and ketamine (Ketalar^®^, 15 mg/kg) were administered intravenously. Thiopental (Pentothal^®^, 25 mg/mL) was administered before tracheal intubation. After oral intubation, for the maintenance of anesthesia, the pig was ventilated (7–8 L/min, 18 rpm) with 40% oxygen in air and isoflurane. As a muscular relaxant, pancuronium bromide (Pavulon^®^, 4 mg/h) was periodically administered [16,17,18].

#### 2.1.2. Placement of Scalp, Subdural and Endovascular Electrodes

Craniotomy was performed after inducing general anesthesia. We provided local anesthesia for the skin incision using lidocaine. Linear dural incisions of about 1.5 cm in length were performed on both hemispheres and 4-contact strip array electrodes (AD-TECH MEDICAL; White Oak, WI, USA) were placed on the surface of each hemisphere and anchored to the peripheral parts of the dura mater (Figure 1). The distance between the center of neighboring contacts was 1 cm.

From the anterior part of the superior sagittal sinus, a 16-gauge peripheral veinous catheter needle was inserted to the sagittal sinus and the outer needle was left in place. This outer needle was anchored with nylon thread. From the outer needle, the electrode of the eEEG was placed in the middle part of the sagittal sinus, parallel to the SDEs.

After placement of the SDEs and eEEG, cranioplasty and skin closure were performed. Using X-rays, we placed the scalp EEG contacts to minimize the distance between the right and left SDE and scalp EEG contacts. We placed another contact on the left tragus as the reference electrode (Figure 2).

To perform scalp, SDE, and eEEG recordings, the sampling frequency was 1 kHz with a bandpass filter of 1 Hz as the low-cut filter and 60 Hz as the high-cut filter.

#### 2.1.3. Swine Animal Model for Focal Cortical Epileptiform Discharges

We performed a 3 mm wide, 3 mm deep corticectomy in the right frontal area, 5 mm lateral to the right SDE, then 1.2 mg of benzyl penicillin diluted with stock solution of 200 units/µL (total volume, 10 µL) was injected into the corticectomy at a depth of 5 mm [15] to artificially create epileptogenicity.

Penicillin is frequently used to induce focal seizures in animal models [19,20,21]. Penicillin is known to induce relative cortical hyperexcitability as a result of decreased inhibitory function of the gamma aminobutyric acid (GABA)-A receptors directly and indirectly due to the beta-lactam effect of penicillin on GABA receptors [22,23]. Penicillin also exerts proconvulsive effects by competition for the benzodiazepine-binding site [24].

We placed the right SDE 5 mm lateral to the artificial epileptogenicity, between Contacts 3 and 4 of the artificially epileptogenic corticectomy (Figure 3).

### 2.2. Endovascular Electroencephalogram (eEEG)

We designed a new eEEG device composed of nickel–titanium alloy wire covered with electrically nonconductive polymer and a platinum alloy electrode, with a length of 3 mm and diameter of 0.2 mm. Compared to conventional microcatheters that can access cerebral blood vessels, this device was an extremely thin, 0.010 inch, guidewire-compatible microcatheter.

### 2.3. Selection of Epileptiform Discharges

#### 2.3.1. Primary Outcome Measurement

We included epileptiform discharges that were: (1) spikes and sharp activities with or without following slow activities; (2) detected by the right SDE only. We excluded epileptiform discharges that were: (1) slow waves or sharply contoured discharges; (2) detected by bilateral SDEs; (3) detected by scalp EEG. Since the eEEG electrode was placed in the superior sagittal sinus, we regarded measurement of epileptiform discharges that were detectable by the left SDE as potentially inappropriate because these discharges could have been secondary synchronized activities through commissural fibers such as the corpus callosum, and we could have, thus, been measuring corpus callosum activities or quasi-generalized activities, rather than the localized epileptogenicity [25]. We recorded scalp, SDE, and eEEG measurements simultaneously for 60 min (Figure 4).

#### 2.3.2. Secondary Outcome Measurement

We also selected epileptiform discharges that had electrical fields identified by a maximum of 3 contacts of the right SDE. Since scalp EEG of the human brain requires >6 cm^2^ of cortical activity [26], compared to 5 cm^2^ for dense array EEG [27] and 4 cm^2^ for MEG [28], we regarded selection of up to 3 contacts of the SDE as appropriate, assuming that the 2 cm maximum distances would cover an area of less than 4 cm^2^. For this measurement, we included sharply contoured activities, because spikes presented less frequently than spike measurements that involved all contacts.

For better understanding, we also measured latency from the peak of eEEG epileptiform discharge to the earliest peak of the right SDE for these localized discharges identified in three or fewer contacts.

## 3. Results

### 3.1. Epileptiform Discharges Detected Only by Right SDE

A total of 636 epileptiform discharges detected by the SDEs were measured during the 60 min recording. Among these, 119 epileptiform discharges met the criteria, indicating that these discharges were localized to the right hemisphere, not the left hemisphere. The eEEG electrode detected 113 of the 119 epileptiform discharges (97%) but missed six discharges (3%). The results, therefore, showed that the sensitivity of the eEEG was 97%. The eEEG did not detect any epileptiform discharges outside those detected by the SDE.

### 3.2. Localized Epileptiform Discharges within Three Contacts of Right SDE

A total of 73 epileptiform discharges was detected that were localized within three contacts (contacts two, three, and four) and did not spread to other parts. Among these, the eEEG electrodes detected 67 epileptiform discharges (92%) and missed six discharges (8%).

The latency between peaks on the eEEG and the earliest epileptiform discharge of the right SDE ranged from 0 ms to 48 ms (mean, 13.3 ms; median, 11 ms; standard deviation, 9.0 ms; standard error, 1.1 ms) (Figure 5).

## 4. Discussion

This study confirmed our hypothesis that an eEEG could detect scalp EEG-undetectable, SDE-detectable interictal epileptiform discharges. What we found in this study was that an eEEG could potentially detect localized epileptogenicity with a 97% sensitivity. The sensitivity of the eEEG in this study was similar to or higher than that described in past studies [12,13,29]. Mikuni et al. performed an eEEG from the cavernous sinus for six patients with mesial temporal lobe epilepsy. Of those six patients, three did not show epileptiform discharges on the scalp EEG, but did on the eEEG [13], which might support our results even though this study independently measured the eEEG and scalp EEG.

Considering the latency [30,31,32,33] and lack of spread to the contralateral hemisphere [34], the epileptiform discharges that the eEEG detected in this study were interictal epileptiform discharges that were at least localized to the right hemisphere. The fact that epileptiform discharges detected by contacts two, three, and four, but not contact one, were detected by the eEEG meant that the eEEG measured activities spreading from electrical fields nearer to contacts two, three, and four of the SDE to the area medial to the location of the eEEG electrode. Considering these results, the eEEG contacts located close to an epileptogenic focus might measure interictal epileptiform discharges with a higher spatial and temporal resolution than a dense array EEG [27] or MEG [28], as the distance between contacts was less than 2 cm. However, to provide a higher spatial resolution, we need to increase the number of contacts in the eEEG with a closer proximity to the epileptogenic focus. Since this study was performed under general anesthesia, we cannot rule out a possible influence of the general anesthesia. These factors, therefore, represent the limitations of this study.

The largest obstacle to accessing the epileptogenic zone from outside the brain is the skull. Therefore, even an SEEG, which is considered to use a less-invasive procedure that is safer than the conventional invasive method, [8] the procedure still requires skull drilling, bolt-anchoring to the skull (some facilities require complete shaving of hair) and puncture to the brain parenchyma. Of course, to be able to validly say this, a study comparing an eEEG and SEEG in terms of the efficacy, safety, patient satisfaction, or preference is required. Our current opinion is that the eEEG could be included as a part of a presurgical algorithm alongside noninvasive studies such as an EEG, video EEG, dense array EEG, MEG, MRI, nuclear imaging, neuropsychological studies and invasive studies such as the SEEG and SDE, and also could be regarded as a “moderately invasive study” [35]. In addition, an eEEG might be useful as a part of a presurgical evaluation.

Since the equipment required for an eEEG in this study was already used not only in endovascular treatments [36,37], but also other treatments [38], we believe that the day will come when this device can be used as a tool in preoperative evaluations for epilepsy surgery.

In future work, we intend to perform an eEEG with the subject in an awake state and to measure not only interictal, but also ictal epileptiform discharges with more contacts located closer to the epileptogenic focus. Furthermore, the newly designed eEEG device used in the present study is an extremely thin, guidewire-compatible microcatheter that is able to easily access cerebral blood vessels. Given its small size, it can likely reach the peripheral blood vessels and could be used in future applications for humans.

## 5. Conclusions

In an anesthetized pig, an eEEG could detect epileptiform discharges that an SDE could detect, but that a scalp EEG could not.

## Figures and Tables

**Figure 1 brainsci-12-00309-f001:**
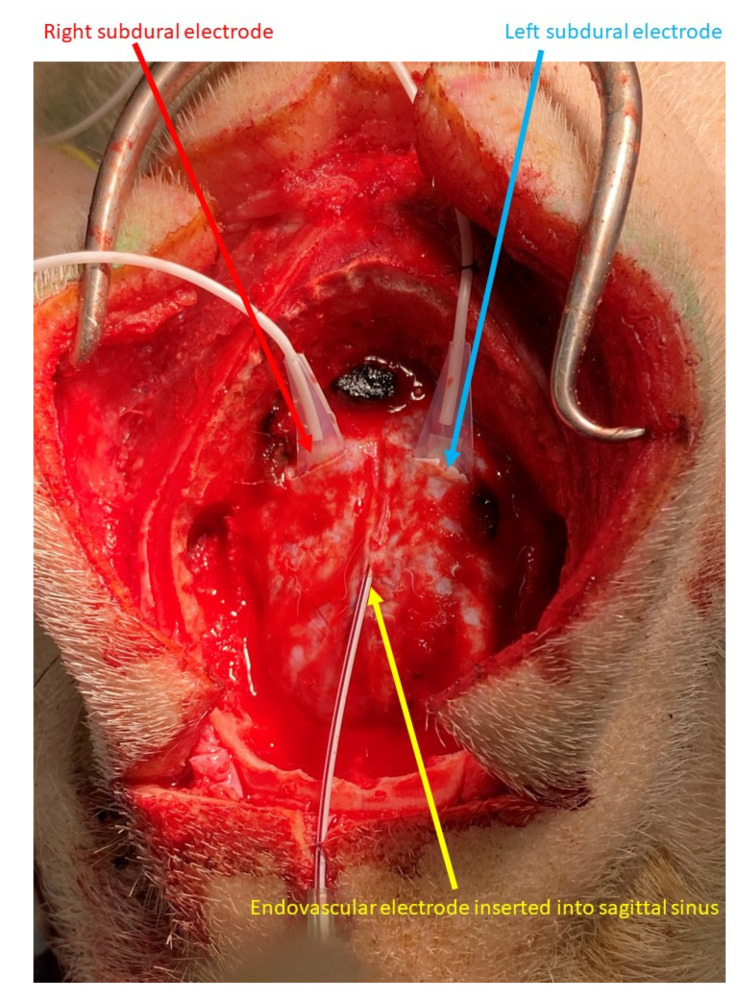
Endovascular and subdural electroencephalograms. Four-contact subdural electrodes were placed in each hemisphere through a linear dural incision (blue and red arrows). The endovascular electrode was inserted into the sagittal sinus through the outer needle of the venous catheter needle (yellow arrow).

**Figure 2 brainsci-12-00309-f002:**
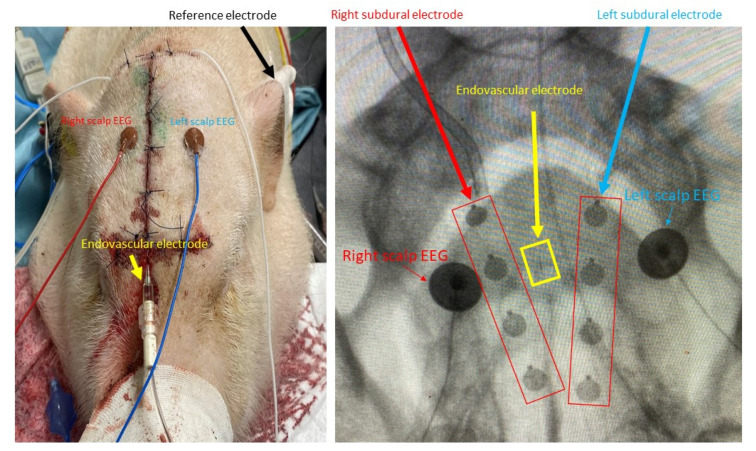
Placement of scalp, subdural, and endovascular electrodes. Each electrode was placed within a close distance using X-ray guidance.

**Figure 3 brainsci-12-00309-f003:**
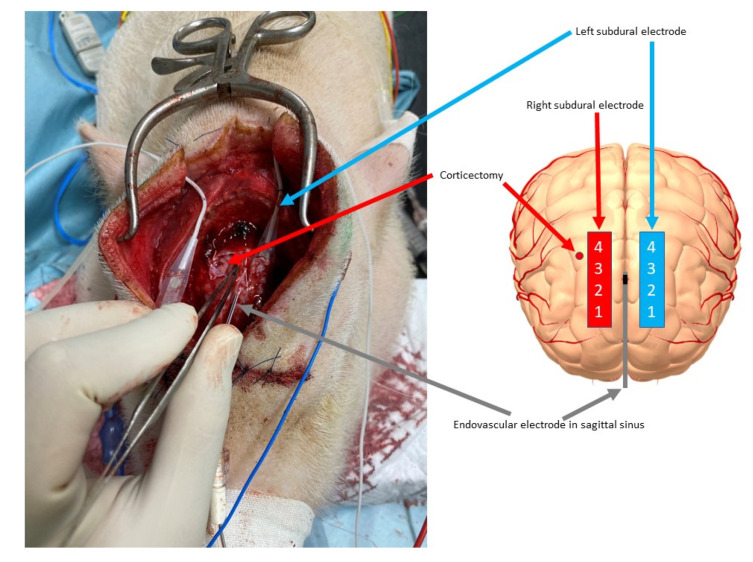
Corticectomy for placement of the epileptogenic focus and positioning of the subdural and endovascular electrodes. On the right frontal area, corticectomy for the epileptogenic focus was performed, 5 mm lateral to the epileptogenic focus and between contacts 3 and 4 of the right subdural electrode (red arrow). The left subdural electrode (sky blue arrow) and endovascular electrode (gray arrow) were already placed (**left**). The location of each electrode and the artificially created epileptogenic area are shown schematically (**right**).

**Figure 4 brainsci-12-00309-f004:**
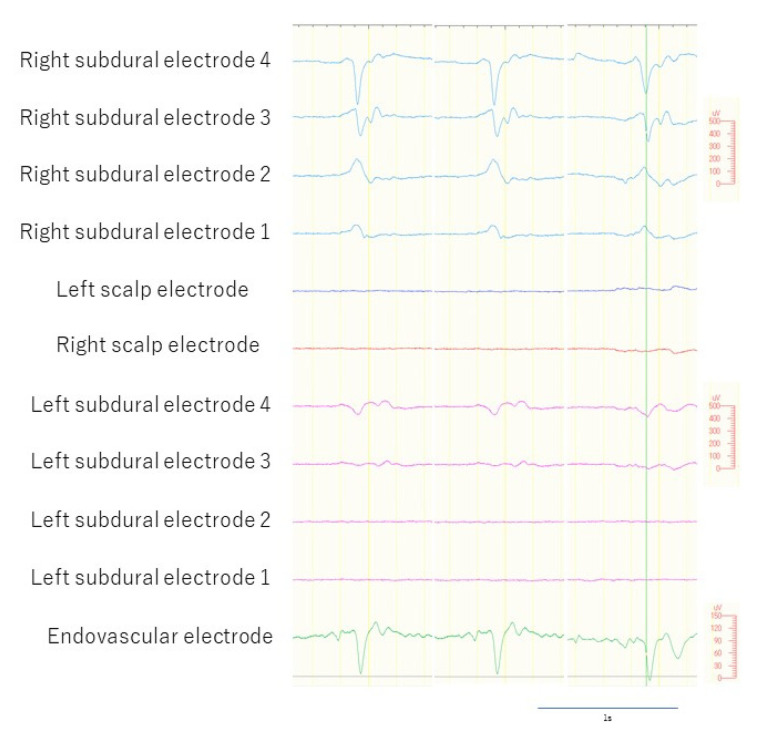
Selected epileptiform discharges. Epileptiform discharges detected by right subdural electrodes (sky blue lines), but undetected by left subdural electrodes (pink lines). The endovascular electrodes detected epileptiform discharges (green line).

**Figure 5 brainsci-12-00309-f005:**
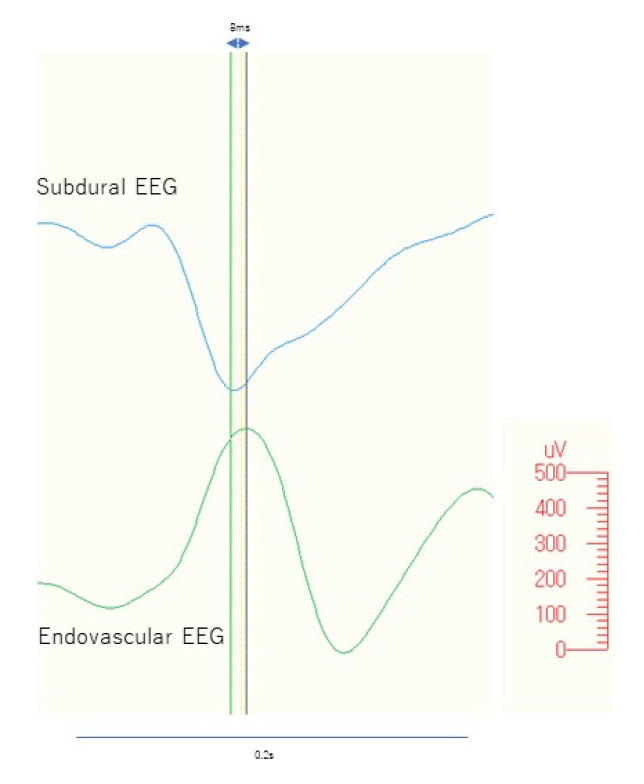
Comparison between subdural and endovascular discharge. Peak-to-peak latency between the endovascular and subdural discharges from one of the representative discharges is shown at 8 ms.

## Data Availability

Data supporting the findings of this study are available from the first author (AF), upon reasonable request.

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
