# Peer review of "Endovascular Electroencephalogram Records Simultaneous Subdural Electrode-Detectable, Scalp Electrode-Undetectable Interictal Epileptiform Discharges"

_brainsci, 2022, doi:10.3390/brainsci12030309_

Round 1
Reviewer 1 Report
In this piece of work, Fujimoto and colleagues performed simultaneous recordings of endovascular EEG (eEEG) with subdural EEG (SE) and scalp EEG in a pig under general anesthesia. They managed to record an artificially generated epileptic field by eEEG that was able to be detected by the SE but not scalp EEG. They conclude that this approach confirmed their hypothesis that eEEG could detect scalp EEG-undetectable, SE-detectable interictal epileptiform discharges.
The paper is nicely written with the exception of the introduction, the material and methods are described in detail, and the conclusions of the study are justified by the findings. I have the following comments:
The introduction of the manuscript is long and repetitive with unnecessarily detailed information about epilepsy. Most importantly, the sixth paragraph contains inaccurate or confusing information. Conventional EEG, dense array EEG, and MEG are electrophysiological measures that have significant clinical utility (see Tamilia et al., Clinical Neurophysiology 2019). MRI is also a neuroimaging technique that provides critical anatomical information about underlying anatomical lesions that may cause epilepsy (see Knake et al., Neurology 2005 and Ryvlin et al., Lancet Neurol. 2014). Although I understand the point that all these techniques are non-invasive compared to the invasive ones, such as stereotaxic EEG, this should be explicitly written in a clearer way. You cannot compare the intracranial EEG to the MRI, since they provide totally different kinds of information. Please re-write this section using the aforementioned references.
The next paragraph starts with the following statement: “Throughout past studies into eEEG,…” Please add the proper references to support this.
The same paragraph states: …but not scalp EEG can be… Please rephrase to …but not scalp EEG may be…
Material and Methods:
It is unclear to me how the artificially epileptic fields were generated. Please provide a more detailed and clear explanation of this.
I would appreciate a comment from the researchers in the discussion on the clinical utility of the endovascular EEG as a presurgical evaluation tool. Stereotaxic EEG dominates now the field of intracranial EEG recordings. How do they believe this technology (i.e., eEEG) compares to the sEEG in clinical practice nowadays and in the near future?
Author Response
Responses to Comments from Reviewer 1
Thank you very much for reviewing our manuscript. We greatly appreciate the advice provided.
In this piece of work, Fujimoto and colleagues performed simultaneous recordings of endovascular EEG (eEEG) with subdural EEG (SE) and scalp EEG in a pig under general anesthesia. They managed to record an artificially generated epileptic field by eEEG that was able to be detected by the SE but not scalp EEG. They conclude that this approach confirmed their hypothesis that eEEG could detect scalp EEG-undetectable, SE-detectable interictal epileptiform discharges.
The paper is nicely written with the exception of the introduction, the material and methods are described in detail, and the conclusions of the study are justified by the findings. I have the following comments:
#1. The introduction of the manuscript is long and repetitive with unnecessarily detailed information about epilepsy.
Response:
We agree. As a matter of fact, there is a history of artificially inflating word counts in order to meet the requirements of the instructions for authors. Therefore, although the text is redundant, we have reverted the text to the original version for the sake of simplicity.
Instead, please understand that the Methods section will be lengthened to meet the requirements of the Instructions along with the revision of the Introduction section.
#2. Most importantly, the sixth paragraph contains inaccurate or confusing information. Conventional EEG, dense array EEG, and MEG are electrophysiological measures that have significant clinical utility (see Tamilia et al., Clinical Neurophysiology 2019). MRI is also a neuroimaging technique that provides critical anatomical information about underlying anatomical lesions that may cause epilepsy (see Knake et al., Neurology 2005 and Ryvlin et al., Lancet Neurol. 2014). Although I understand the point that all these techniques are non-invasive compared to the invasive ones, such as stereotaxic EEG, this should be explicitly written in a clearer way. You cannot compare the intracranial EEG to the MRI, since they provide totally different kinds of information. Please re-write this section using the aforementioned references.
Response:
We are sorry for any confusion. We have rewritten this paragraph in accordance with the advice provided, citing the suggested references.
#3. The next paragraph starts with the following statement: “Throughout past studies into eEEG,…” Please add the proper references to support this.
Response:
Thank you very much for pointing out this oversight. We have added a reference to part research.
#4. The same paragraph states: …but not scalp EEG can be… Please rephrase to …but not scalp EEG may be…
Response:
We have removed this part, which had already been explained.
Material and Methods:
It is unclear to me how the artificially epileptic fields were generated. Please provide a more detailed and clear explanation of this.
Response:
We have added further explanation about this issue.
I would appreciate a comment from the researchers in the discussion on the clinical utility of the endovascular EEG as a presurgical evaluation tool. Stereotaxic EEG dominates now the field of intracranial EEG recordings. How do they believe this technology (i.e., eEEG) compares to the sEEG in clinical practice nowadays and in the near future?
Response:
Thank you very much for these comments. We appreciate this and have provided some more comments on the future utility of eEEG.

Reviewer 2 Report
Very well designed study showing proof of concept for eEEG. Adding a few lines about the risks and relative benefits of eEEG would improve the paper but are not necessary. You can consider adding images of epileptiform discharges on subdural and endovascular EEG for comparison as well.
Author Response
Responses to Comments from Reviewer 2
Thank you very much for reviewing our manuscript. We greatly appreciate the advice provided.
#1. Very well designed study showing proof of concept for eEEG. Adding a few lines about the risks and relative benefits of eEEG would improve the paper but are not necessary.
Response:
Thank you very much for the comments. Since this is the first animal experiment involving eEEG, please give us more time to state the risks of the eEEG. However, we would like to add the possible utility of eEEG as a benefit of the tool in the Discussion section.
You can consider adding images of epileptiform discharges on subdural and endovascular EEG for comparison as well.
Response:
Thank you very much for this suggestion. Even though a comparison of discharges between SDE and eEEG has already been shown in Figure 4, we are sorry for the confusion. To facilitate reader understanding, we have added a new figure, as Figure 5.
